



# Deep-sea ecosystem: a world of positive biodiversity – ecosystem functioning relationships?

Elisa Baldrighi[1], Donato Giovannelli [1,2,3], Giuseppe d'Errico [4], Marc Lavaleye[5] and Elena Manini[1]

*Correspondence to*: E. Baldrighi (elisa.baldrighi@an.ismar.cnr.it)
                E. Manini (e.manini@ismar.cnr.it)

[1]Institute of Marine Sciences – ISMAR, National Research Council – CNR, 60125 Ancona, Italy
[2]Institute of Earth, Ocean and Atmospheric Science – EOAS, Rutgers University, New Brunswick, 08901 NJ, USA
[3]Institute of Advanced Studies, Program in Interdisciplinary Studies, 08540 Princeton, USA
[4]Department for Life and Environmental Science – DISVA, Polytechnic University of Marche – UNIVPM, 60131 Ancona, Italy
[5]Department of Marine Ecology, Royal Netherlands Institute for Sea Research (NIOZ), Texel, The Netherlands



**Abstract**: The global scale of the biodiversity crisis has stimulated research on the relationship between biodiversity and ecosystem functioning (BEF) in several ecosystems of the world. Even though the deep-sea seafloor is the largest biome on Earth, BEF studies in deep-sea benthic ecosystems are scarce. In addition, the few recent studies, mostly focus on meiobenthic nematodes, report quite different results spanning from a very clear positive relationship to none at all. If deep-sea BEF relationships are indeed so variable or have a more common nature is not established. In this first BEF study of deep-sea macrobenthic fauna, we investigated the structural and functional diversity of macrofauna assemblages at three depths (1200, 1900 and 3000m) in seven different open slope systems in the NE Atlantic Ocean (n=1) and Western (n=3) and Central (n=3) Mediterranean Sea. The results demonstrate a positive relationship between deep-sea macrobenthic diversity and ecosystem function, with some variability in its strength between slope areas and in relation to the spatial scale of investigation and environmental conditions. The macrofauna functional diversity did not appear to be more effective than structural diversity in influencing ecosystem processes. Rare macrofaunal species were seen to have a negligible effect on BEF relationship, suggesting a high ecological redundancy and a small role of rare species in providing community services.



# 1 Introduction

Earth is experiencing a pervasive and uncontrolled loss of species, which has raised concerns about the deterioration of ecosystem functions and services (Gagic et al., 2015). This scenario has stimulated research that helps to understand the biodiversity-ecosystem function relationships (BEF), to clarify how ecosystems work and respond to change, and if and how biodiversity matters (Loreau, 2010). A large body of studies dealing with BEF relationships have been produced during the past decades and reviewed in recent literature (Cardinale et al., 2011; Tilman et al., 2014). Most of the research has been conducted in terrestrial environments or in the laboratory, where manipulative experiments can be performed under controlled conditions. Despite the number of studies in marine systems has increased rapidly over the past few years (Worm et al., 2006; Mora et al., 2011), only recently BEF was also studied in the deep-sea (>200 m depth; 7-9). BEF research in many terrestrial, freshwater and marine systems (Tilman et al., 2014) has clearly shown that biodiversity affects the ecosystem functioning. The relationship is overall positive, especially in complex systems and over long timescales (Cardinale et al., 2007). Such positive effect is often related to the fact that different animals have complementary functions, rather than competitive. Furthermore, it has also been suggested that functional diversity, rather than species richness, was a better predictor of the ecosystem functioning (Cardinale et al., 2011; Lefcheck and Duffy, 2014). However, not all studies report the same general trend, and conflicting results have been described in small-scale experiments and complex, large-scale observations (Lefcheck and Duffy, 2014). New studies carried out under natural conditions across spatial and temporal scales and under the influence of different environmental conditions are necessary (Gamfeldt et al., 2014; Snelgrove et al., 2014).

Seafloor environments cover over 70% of the Earth surface, and significantly contribute to global ecosystem functions and services (Snelgrove et al., 2014). The deep-sea is the most extensive and highly diversified environment on the planet, and provides the main long-term sink for carbon flux from the photic zone (Gage and Tyler, 1991). Animals such as nematodes (Ingels and Vanreusel, 2013) and burrowing invertebrates (Lohrer et al., 2004) modify the seafloor habitat for microbes, significantly altering carbon flux, storage, and recycling nutrients over multiple timescales (Lohrer et al., 2004), thus playing an important role in the global ecosystem. Assessing the effect of a further and possible loss of biodiversity on ecosystem functioning due for instance to global warming, is thus of the utmost importance (Norkko et al., 2006). BEF relationships previously reported for deep-sea benthic communities (Danovaro et al., 2008) show: i) a prevalence of mutualistic interactions between organisms rather than competition interactions (Loreau, 2008) in different deep-sea habitats at different longitudes and latitudes; and ii) the loss of species can





seriously affect the ecosystem functioning in a negative way (Danovaro et al., 2008). However, the
strength of BEF relationships can differ strongly from habitat to habitat (Lefcheck and Duffy, 2014;
Thurber et al., 2014), in relation to the functional traits and the species involved (O'Connor and
Crowe, 2005). For example, a study (8) performed in open slope systems reported that BEF
relationships are non existent. Most deep-sea BEF investigations have used the meiofauna,
particularly nematodes, as model taxon (Danovaro et al., 2008; Leduc et al., 2013; Pape et al., 2013;
Pusceddu et al., 2014a, 2014b), whereas comparatively few studies have examined the role of
microbial and viral components (Brandt, 2008; Glud et al., 2013) or of larger epifauna (Amaro et
al., 2013), in enhancing ecosystem functioning. In this study we investigate the BEF relationships in
the deep-sea by considering the structural and functional diversity of macrobenthic fauna.
Macrobenthos is recognized to have important ecological roles, namely in bioturbation (Loreau,
2008), sediment oxygenation, and as an important food source for higher trophic levels (Gage and
Tyler, 1991). Macrobenthos has been largely used for shallow-water and freshwater BEF
investigations (Gamfeldt et al., 2014; Lefcheck and Duffy, 2014) but, to the best of our knowledge,
not yet for deep-sea BEF studies. Since setting up *in situ* experiments in the deep-sea is difficult and
costly, we used the observational - correlative approach, to test the truth of the following three
hypotheses: 1) functional diversity affects ecosystem functioning more than species richness,
(Naeem et al., 1994); 2) the spatial scale of investigation and related environmental factors, affect
the findings of BEF studies; and 3) the number of 'rare' species, supposed to be the vast majority of
species in the deep-sea (Gaston, 1994), affects the nature and strength of BEF relationships.

**2 Material and Methods**
**2.1. Study area**
Seven open slopes positioned along a west-east axis from the NE Atlantic Ocean to the Central-
Eastern Mediterranean basin were selected for the study (Fig. 1). The Atlantic sampling area was on
the Galicia Bank, a seamount situated on the Iberian margin about 200 km off the Galician coast.
The Galicia Bank, with a summit at 620 water depth, is separated from the shallower parts of the
continental margin by the Galicia Interior basin (Pape et al., 2013), which has an approximate depth
of 3000 m (Fig. 1). Waters current velocities registered on top of the seamount are 5–30 cm s-1
(Pape et al., 2013), and are high enough to influence organic matter deposition. This in turn results
in very low concentrations of phytopigments and biopolymeric organic carbon at 1200 m depth on
the seamount (Table S1), along with the presence of coarse sediments (Table S1). The deep
Mediterranean Sea is a highly oligotrophic environment (Giovannelli et al., 2013), characterised by
a well-established trophic difference between the more productive western and the less productive



eastern basin (Baldrighi et al., 2014). The gradient is generated by higher nutrient input in the
western Mediterranean Sea due to river runoff, the inflow of Atlantic surface water, and the outflow
of relatively nutrient-rich Levantine Intermediate Water through the Strait of Gibraltar (Bergamasco
and Malanotte-Rizzoli, 2010).

**2.2 Sampling strategy**
Biological and environmental samples were collected during several cruises in the framework of the
BIOFUN project ('*Biodiversity and Ecosystem Functioning in Contrasting Southern European*
*Deep-sea Environments: from viruses to Megafauna*'). Sediment samples were collected from the
seven open-slope areas: one in the NE Atlantic (ATL), three in the Western Mediterranean basin
(wM1, wM2 and wM3) and three in the Central-Eastern Mediterranean basin (c-eM1, c-eM2 and c-
eM3) (Fig. 1). All of the selected open-slope systems in the Mediterranean Sea were from
topographically regular settings and characterized by different trophic and oceanographic conditions
(D'Ortenzio et al., 2009; Giovannelli et al., 2013) (Table S1). At each slope, three stations at three
different depth ranges were sampled and namely: upper bathyal (1,200 m), mid-bathyal (from
1,800 to 1,900 m), and lower bathyal (from 2,400 to 3,000 m). c-eM1 could not be sampled at the
lower bathyal depth range: this station was substituted with another at 2,120 m (Table S1). At each
station, independent replicate samples (n=3) were collected to analyse macrobenthos, meiobenthos,
microbial component and environmental variables using a cylindrical box-corer (internal diameter
32 cm, except for ATL, wM2 and c-eM2 areas where the internal diameter was 50 cm).

**2.3 Environmental and biological sampling**
To analyse grain size, biochemical composition of the organic matter and microbial component,
subsamples of the sediment from each box-corer were collected using plexiglas cores with an
internal diameter of 3.6 cm. The top 1 cm of one subcore of each box corer was collected and
frozen at −20 °C, for the analysis of chlorophyll-a, phaeopigment and organic matter content.
Replicates of about 1 ml wet sediment were fixed using buffered formaldehyde and stored at + 4 °C
until processing for total prokaryotic abundance and biomass determination (Giovannelli et al.,
2013). The top 20 cm were preserved at + 4 °C for grain size analysis. For meiofauna analysis,
sediment was taken from each box corer using a plexiglas tube with an internal diameter of 3.6 cm
and immediately fixed in 4 % buffered formalin and Rose Bengal; once in the laboratory, only the
top 5 cm was sieved through a 300 μm and 20 μm mesh sieve. Meiofaunal samples were collected
only from six of the seven selected areas. For macrofauna analysis, the top 20 cm of sediment from



each box-corer sample, along with their overlying water, was sieved through a 300 μm mesh sieve
to retain all the macrobenthic organisms (considered here as *sensu lato,* as reported in Baldrighi and
Manini (2015). The residue left behind on the sieve was immediately fixed in buffered formalin
solution (10 %), and stained with Rose Bengal.

**2.4  Environmental and faunal samples processing**
Grain size, phytopigment contents, quantity and biochemical composition of organic matter
analyses were performed as reported in Baldrighi et al. (2014). Total prokaryotic number and
biomass were estimated as reported in Giovannelli et al. (2013). Meiofauna abundance, biomass and
diversity estimation were analysed according to Baldrighi and Manini (2015). Macrofauna
abundance, biomass and biodiversity analyses has been described in detail by Baldrighi et al.

199  (2014).


**2.5  Macrofaunal biodiversity and functional diversity**
For each slope, we analysed  the macrobenthic community diversity and functional traits (Table
S2). Macrobenthic organisms were counted and classified to the lowest possible taxonomic level.
Biodiversity was measured as richness of macrofauna higher taxa (n° taxa), species richness (SR),
or total number of species collected in each box corer sample and the expected number of species
$ES_{(n)}$ for theoretical samples of  n = 50 individuals. This last method of rarefaction provides a good
tool for comparisons of species richness among samples that have different total abundances or
surface areas (Danovaro et al., 2008). Functional diversity is the range of functions that are
performed by the organisms in a system (Cardinale et al., 2011). We used four different indices as
proxies for the functional diversity of the macrofauna: 1) trophic diversity ($\Theta^{-1}$); 2) the expected
number of deposit feeders ($EDF_{(30)}$), 3) the expected number of predator species ($EPR_{(20)}$); and 4)
the bioturbation potential estimation (BP) (Baldrighi and Manini, 2015; Quéiros et al., 2015). Given
that micro- and meiofauna are both affected by environmental changes, particularly those generated
by bioturbation by the macrofauna (Piot et al., 2013), we considered the effect of the BP on
prokaryotic and meiofaunal biomass. The presence of 'rare' species in samples was estimated based
on two definitions of rarity (Cao et al., 1998): singleton (i.e., species with an abundance of one in
one sample) and 'rarest of rare' (i.e. species occurring with an abundance of one in single sample in
the entire dataset).




### 2.6 Ecosystem functioning and efficiency

Deep sea ecosystem functioning was estimated as benthic faunal biomass (mgC m$^{-2}$; Danovaro, 2012) considering total benthic biomass (the sum of prokaryotic, meiofauna and macrofauna biomass) and the biomass of the functional group of macrobenthic predators (Table S3). Biomass is a measure of the production of renewable resources by an ecosystem (Rowe et al., 2008) and a reduction in the predator population size may exert effects that go beyond top-down control, thus affecting cross-system connectivity and ecosystem stability (McCauley et al., 2015). To measure the ecosystem efficiency three indicators were used: i) the ratio of macrofaunal biomass to biopolymeric carbon content (MBM:BPC), which is an estimate of the ability of the system to channel detritus to higher trophic levels (Danovaro, 2012); ii) the ratio of macrofaunal biomass to prokaryotic biomass (MBM:TPB); and iii) the ratio of macrofaunal biomass to meiofaunal biomass (MBM: MEB). A large number of deep-sea macrobenthic organisms are identified as deposit feeders, which ingest large amounts of sediment with detritus, prokaryotes and meiofauna (Baldrighi and Manini, 2015). It has been suggested (Van Oevelen et al., 2006) that up to 24 % of total bacterial production is grazed by macrofauna, and that meiofauna is an important link between smaller (e.g., bacteria) and larger organisms (e.g., macrofauna). The MBM: TPB and MBM: MEB ratios are thus measures of the energy transfer from lower to higher trophic levels based on the hypothesis that macrofauna predates on microbial and meiofauna components.

### 2.7 Statistical analysis

BEF relations can be determined by the effect of the spatial scale of investigation and environmental factors that act at each scale (Gamfeldt et al., 2014). We investigated the presence of BEF relations considering: i) a large spatial scale, encompassing our entire dataset (i.e. all data of our three research areas were taken together during the statistical analysis) and ii) a basin spatial scale, where the data of the three different sampling areas (NE Atlantic Ocean, Western and Central-Eastern Mediterranean basins) were kept separate during the statistical analysis. The relationships between BEF and efficiency were estimated by a linear model (in the form y = a+bx), a power model (y = a+x$^b$) and an exponential model (y = e$^{a+bx}$). Linear, power and exponential models are currently considered as the best tools to describe BEF relationships in different deep-sea environments (Cardinale et al., 2007; Danovaro et al., 2008; Lefcheck and Duffy, 2014). Statistical analyses were performed using R-cran software (http://www.R-project.org). Map plots were drawn using Ocean Data View (Schlitzer, 2011). Relationships between variables were tested using linear and non-linear regression. After fitting the 3 models to the experimental data, the distribution of the residuals, r$^2$ and the Akaike Information Criterion (Akaike, 1974) were used to discriminate the best



fitting model, as appropriate. Model fitting was performed for two spatial scales, large scale, i.e. the
entire dataset, and basin scale, i.e. the sampling area (Atlantic Ocean, wM basin, and c-eM basin).
Distance-based multivariate regression analysis with forward selection (DISTLM) (Anderson,
2004) was used to account for the potential effect of environmental features on BEF relationships.
The effects of depth, longitude, temperature, grain size, quantity and quality of food sources were
included as covariates in the analyses. P values were obtained with 4999 permutations of residuals
under the reduced model.

**3 Results and Discussion**
**3.1 Large sampling spatial scale hinders the identification of BEF relationships**
Continental slopes are valuable sites for investigations of BEF relationships. They account for more
than 20% of total marine productivity, and for a significantly greater proportion of organic matter
exports to the seafloor. Slope sediments host a large proportion of marine biodiversity and are
repositories of deep-sea biomass (Baldrighi et al., 2013). The large spatial scale data (i.e. from the
Atlantic Ocean to the Central-Eastern Mediterranean Sea) show that macrofauna diversity (SR) was
significantly and exponentially related to ecosystem functioning (Fig. 2a, Table S4). An
exponential relation between biodiversity and ecosystem functioning has been previously reported
for various organism size classes (Mora et al., 2014). Positive interspecific interactions between
organisms, such as facilitation, have been suggested to sustain such relations (Danovaro et al.,
2008). In the present study, not all the diversity indices used were significantly related to the
ecosystem functioning measures (Table S5a). Actually, the existence of a BEF relationship
appeared to be closely linked to the diversity and ecosystem functioning measures used (Gamfeldt
et al., 2014), which are often context-dependent (O'Connor and Donohue, 2013). SR was the only
diversity index positively relationship with total benthic biomass, while $ES_{(50)}$ was related to
macrobenthic predator biomass (Table S4). The relationships between other diversity indices and
benthic biomass were explained by the environmental cofactors (water depth, longitude, food
availability and grain size). These data are in line with other studies (Pape et al., 2013; Cusson et
al., 2015; Poorter et al., 2015) where not all diversity measures correlated with ecosystem
functioning.
The positive influence of biodiversity on ecosystem efficiency, can be understood if we suppose
that with a high biodiversity most niches within an ecosystem are filled, whereby the available food
sources can be used very efficiently, and be converted into a higher biomass (Naeem et al., 1994).
For the quantification of energy flow through the biotic ecosystem we use the ratio between
macrobenthic biomass and the amount of biopolymeric carbon as a proxy. This ratio between





macrobenthic biomass and biopolymeric carbon was previously suggested to be a proxy for
ecosystem efficiency (Danovaro, 2012), even though it has been reported to have both a positive
relationship (Danovaro et al., 2008) and no relationship with benthic diversity (Leduc et al., 2013).
The quantification of energy flow through the ecosystem by using the ratio between macrobenthic
and microbial biomass or between macro- and meiofaunal biomass are other proxies for how
efficiently the ecosystem works (Cardinale et al., 2012); the higher the two ratios, the more efficient
the system. However, this is a gross simplification of the energy flow through an ecosystem, as this
will be rarely a direct flow from the smaller to the bigger organisms but is much more complicated
and will be influenced by many biotic interactions (Piot et al., 2013) and abiotic variables
(Snelgrove et al., 2014; Tilman et al., 2014). In the present study, macrobenthic biodiversity was
not significantly related to the three ecosystem efficiency proxies. Most of ecosystem efficiency
variability was explained by environmental covariates (Table S4a).
Macrofauna functional diversity was expressed as trophic diversity, *i.e.* EDF$_{(30)}$, EPR$_{(20)}$, and BP.
BEF relationship was found only when EDF$_{(30)}$ was considered, and it was significant and
exponential (Fig. 2b, Table S4). Deposit feeders were the most abundant trophic group, suggesting a
key role for them in ecosystem functioning. None of the other functional diversity indices used had
any effect on ecosystem functioning, or else the relationships were explained by a covariate effect
(Table S5a). There was no relationship between EPR$_{(20)}$ and total biomass, but only a slightly
positive trend; indeed, higher numbers of predator species did not correlate with higher biomass
values. Moreover, there was no correlation between the predator number (ind/m$^2$) and their biomass
(R$^2$= 0.03, p> 0.05). In particular, the wM slope systems were characterized by a high number of
predators and a high EPR$_{(20)}$ while their biomass values were lower than those measured in the
Atlantic slope area. This dwarfism of macrobenthic organisms inhabiting the Mediterranean Sea
compared with Atlantic Ocean, is well established (Baldrighi et al., 2014). Bioturbation activity of
organisms can affect both the abiotic and biotic components of a system (Quéiros et al., 2015) and
has been identified as one of the functional traits of benthic organisms that may sustain mutualistic
interactions on the basis of BEF relationships (Loreau, 2008). On the large spatial scale,
bioturbation was the only functional parameter that is positively and linear correlated with
ecosystem efficiency in terms of the MBM : BPC ratio (Table S4). This finding supports the idea
that bioturbation can facilitate organic matter recycling and its uptake by higher trophic levels
(Quéiros et al., 2015). The linear relation indicates that all organisms contribute to similar extents to
ecosystem efficiency (Naeem et al., 1995). In all the other cases (Table S5a), the bioturbation effect
on ecosystem functioning and efficiency was overridden by covariate effects. The mutually positive
functional interactions among macrobenthic organisms may explain the exponential nature of the
BEF relationships detected (Danovaro, 2012). It is also conceivable that competitive displacement,





exclusion and/or predation, interactions that usually occur in shallow water hard substrate systems,
are weak in soft sediment, where direct competition for space and food rarely plays important role
(Gage, 2004). In the deep sea, the generally low density of organisms would further weaken any
interaction between species (Gage, 2004). Indeed, in the deep sea a predominance of mutualistic
interactions is more conceivable than competition or even a saturation effect (Gage, 2004).
Nevertheless, the effect of environmental variables affected many of the BEF relationships detected.
The steep environmental gradients characterizing the Atlantic – c-eM transect can easily influence
BEF relationships on large scale (Cusson et al., 2015). Contrary to expectations, the functional
diversity indices used did not explain ecosystem functioning more exhaustively than the traditional
biodiversity indices. This suggests that they may not encompass the full array of key macrobenthic
functional traits that underpin ecosystem functioning and efficiency processes. According to recent
studies, isotopic analysis can be a promising tool to clarify trophic niches (Rigolet et al., 2015). The
present findings also show that the effect of functional diversity on ecosystem functioning is closed
related to the spatial scale considered and that taxonomic and structural attributes as well as
ecosystem properties and processes may vary along environmental gradients.

## 3.2 Disentangling BEF relationships on the basin spatial scale

It has been hypothesized that BEF relationships are spatial scale- and context-dependent, and that
their nature is related to the system analysed and the organisms involved (Ieno et al., 2006; Poorter
et al., 2015). The environmental context appeared to be determinant also in our study, where a
different situation was found in each of the three slope systems (Table S6). In the w-M basin
macrofauna diversity showed a clear, positive relation with ecosystem function and efficiency (Fig.
3, Table S6), whereas in the other areas (Table S5b) the effect of environmental variables attenuated
the BEF relations. The nature of these relationships ranged from linear to exponential, according to
the proxies that were applied to quantify biodiversity. However, independently from the nature of
the relationships, macrofauna diversity in the w-M basin has a positive effect on ecosystem
functioning and efficiency. As regards macrofaunal functional diversity, a highly significant and
exponential relationship was detected between $EPR_{(20)}$, $EDF_{(30)}$ and ecosystem functioning in the w-
M basin and in the Atlantic area (Fig. 4a, b and c, Table S6), but not in the c-eM basin. With respect
to the relationships between functional diversity and ecosystem efficiency, macrofauna functional
diversity exhibited an exponential relationships to one of the proxies of ecosystem efficiency (i.e.
MBM : MEB ratio) (Table S6) whereas a null relation was found for the Atlantic area, and the
relation was mostly explained by the effect of environmental factors in the c-e M basin (Table
S5b).



Taken together, the present findings confirm that environmental drivers, SR and functional diversity affect ecosystem functioning in different ways and with different strength, based on spatial scale (Cardinale et al., 2007; Poorter et al., 2015). Indeed, some BEF relationships that were highlighted on the basin scale were not appreciable on the larger scale, probably due to masking effects exerted by environmental features.

Such effects were very strong in the c-eM basin, where most relationships were context-dependent (Table S5b). In the Eastern Mediterranean basin the environmental conditions, such as food depletion or current regime have been reported to be major factors influencing and structuring the benthic populations (Kröncke et al., 2003). According to our data, environmental variables completely governed BEF relationships in this area. Nonetheless, other benthic components, for instance meiobenthic nematodes (Danovaro et al., 2008; Danovaro, 2012), may exhibit different response. As noted by Pusceddu et al. (2014a), the presence and shape of BEF relationships can vary when different components (meiofauna, macrofauna or fish) are taken into account. This suggests that different environment contexts (i.e. basins) may involve considerable change in the functional structure of the macrobenthic communities (e.g., turnover in species composition) (Baldrighi et al., 2014). O'Connor and Crowe (2005) concluded that different species played idiosyncratic roles, explaining why in some cases no relationship can be found between SR and ecosystem functioning. As noted above for large spatial scale analysis, the functional diversity indices used did not explain ecosystem functioning more exhaustively than conventional biodiversity indices, at least for the functional measures that we adopted.

### 3.3 Are rare species driving biodiversity – ecosystem functioning relations?

Previous studies suggested that the deep-sea ecosystem is characterized by the presence of rare species, and that this is as an emergent property of high-diversity systems (Gage, 2004). Key ecosystem processes may be threatened by the loss of species that perform specific functions, some of which may be rare (Mouillot et al., 2013). However, the issue of rare species is still in its infancy and many questions are still open: how do we define rare species? Are rare species a product of sampling size, a taxonomic bias or is it a genuine phenomenon? (Mouillot et al., 2013).

In the present study defined rare species considering two degrees of rarity, rare species defined as 'singletons' and species that were 'rarest of the rare' (see Sect. 2.5). The presence of singletons characterized only two slope areas in the wM basin (wM1 and wM3) at all depths sampled. Their contribution in terms of rare species richness to the total SR was between 24% (wM1 at 2400 m and wM3 at 1200 m) and 45% (wM3 at 2400 m). When $ES_{(50)}$ was computed out of the total number of



expected species, they accounted for a proportion that ranged from 5% (wM3 at 1200 m) to 13%
(wM1 at 1900 m). The contribution of rare species to the total macrofaunal abundance in terms of
abundance ($ind/m^2$) never exceeded 8 %, ranging from 1 % (wM3 at 1200 m and 1900 m) to 8 %
(wM1 at 1900 m). Moreover, the number of rare species did not correlate with the value of total SR
in any slope area. Such a correlation has been reported in some studies (Kerr, 1997; Ellingsen,
2002), but not in others (Schlacher et al., 1998). The set of rare species found in the three open-
slope systems investigated was structurally and functionally similar to the total observed species
pool. Singletons included several taxa (e.g. Annelida, Mollusca, Crustacea, Nematoda, Bryozoa,
Sipuncula) from all four trophic groups considered. In particular, each depth was characterized by a
typical 'singleton community', indicating a quick change in the rare species composition along each
slope area. As reported by Fried et al. (2015) the functional structure of a macrobenthic community
showed less variation than species composition, due to the natural bathymetric zonation
characterizing communities in continental margins (Mouillot et al., 2013). To assess the effect of
singletons on the BEF relationships identified in this study (see Table S6), rare species were
removed from the dataset and all diversity and functional diversity indices recomputed. As expected
the $EDF_{(30)}$ and SR values significantly decreased (ANOVA, $p < 0.05$) compared to the original
values (Table S2), however the other indices (i.e. $ES_{(50)}$, Taxa richness, $EPR_{(20)}$, PB) did not changed
significantly. All the significant BEF relations identified both for all studied areas together (i.e.
large spatial scale) as well as for each basin were unaffected in nature and strength by the removal
of rare species. This can be explained by the fact that rare species share a combination of functional
traits with more common species, which would ensures the persistence of a those functional traits at
the ecosystem level even in case of loss of some species (Fonseca and Ganade, 2001). Our findings
are in line with the data reported by Ellingsen et al. (2007) in marine soft sediments from New
Zealand, and suggests a role for rare species in community resilience (Törnroos et al., 2014), and
potentially in providing ecological redundancy in the deep-sea environments (Fonseca and Ganade
,2001). Data analysis demonstrated that some 'singleton' species in a slope area were not rare in
others, probably due to different habitat conditions; this is in line with niche theory, which suggests
that as environmental gradients are crossed, many species should change from being rare to
abundant and *vice versa* (Ellingsen et al., 2007). This finding prompted the adoption of an extreme
definition of rarity: 'rarest of the rare', *i.e.* species occurring with an abundance of one in a single
sample in the entire dataset. The contribution of such species to the total diversity never exceeded 4
% and their abundance was always equal to or less than 1 %. Their effect on BEF relations was
always negligible. Our findings are not in line with the general theory of the huge number of rare
species in the deep-sea and their key role in the system (Gage, 2004). The number of rare species,
however, can be dependent on the sample size. It can be imagined that with a limited number of
species in an area , that the larger the sample the smaller the number of singletons will be and thus
that the appropriate scale to study rare species could be much larger than those usually used for
benthic diversity investigations (Gray, 2002). However, rare species often remain as singletons even
after adding up large numbers of replicates from the same area (Gage, 2004). Moreover, rarity is
often associated with traits related to dispersal ability (Gaston et al., 1997). This consideration
applies to our dataset, because most of our 'rare' species were peracarid crustaceans (e.g.,
*Leptognathia aneristus, Cyclaspis longicaudata, Diastyloides serratus, Eurycope sp.*) that have a
direct development and a much more limited potential for dispersal (Gage, 2004), in contrast to
species with a planktonic larval stage. It is also possible that rare species are widely distributed;
however, their rarity in samples and problems of reliable estimation from such low-density
populations means they have been collected at a single place.

**4 Conclusions**
Taken together the present data demonstrate that the spatial scale of the investigation and related
environmental factors determines the presence and form of the relationship between deep-sea
macrofaunal diversity and ecosystem function and efficiency. Macrofauna biodiversity positively
affects ecosystem functioning. Functional diversity did not seem to be more effective in promoting
ecosystem processes than structural diversity *per se*. At least, their effectiveness changes from basin
to basin and according to the environmental features. The challenge for future studies is to identify
functional traits that affect ecosystem processes in multiple environmental contexts. The issue of
rarity and the effect of rare species on ecosystem processes remains to be explored. Species are rare
for a variety of reasons, including sampling artefacts and genuine rarity (Gaston et al., 1997). Two
main issues need to be addressed: (1) whether rarity is a genuine phenomenon and (2) which key
functional traits of rare species may be crucial in maintaining ecosystem functions. Future BEF
studies should consider the integration of different size classes and trophic levels (e.g. meio- and
macrofauna) to achieve more realistic conclusions, as also noted by Piot and co-authors (2013).
Understanding BEF relationship and underlying processes is critical to preserving the deep-sea
ecosystem and its functioning and is a precondition for its sustainable exploitation.








**Acknowledgments**
The authors are indebted to the crews of the ships R/V Pelagia (The Netherlands), R/V Urania
(Italy) and R/V Meteor (Germany) for their help during the sampling activities. This study is part of
the ESF EuroDEEP project BIOFUN (CTM2007-28739-E) and writing of the manuscript was
supported by BALMAS (IPA ADRIATIC project; 1uSTR/0005). DG was supported by a C-DEBI
(Center for Dark Energy Biosphere Investigation) postdoctoral fellowship.

**Author contributions:** E. Baldrighi and E. Manini designed research; E. Baldrighi performed
research; E.Baldrighi, G. d'Errico and D. Giovannelli analyzed data; E. Baldrighi prepared the
manuscript with contributions from all co-authors.




















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

## Figure Legends

**Fig. 1.** Map of the study area and sampling sites. Purple circle, Galicia bank - Atlantic ocean (ATL); red circles, Western Mediterranean basin (wM1, 2, 3), yellow circles, Central-Eastern Mediterranean basin (c-eM1, 2, 3).

**Fig. 2.** Large spatial-scale relationships between macrofauna biodiversity and ecosystem functioning and efficiency. (a) Relationship between species richness (SR) and ecosystem functioning expressed as total benthic biomass (mgC/m$^2$). The equation of the fitting line is $y = e^{(-1.08+0.13x)}$ (N= 64; $R^2 = 0.98$; P< 0.001). (b) Relationship between functional diversity, expressed as expected richness of deposit feeders (EDF$_{(30)}$), and ecosystem functioning (total benthic biomass). The equation of the fitting line is $y = e^{(2.64-0.16x)}$ (N=64; $R^2 = 0.89$; P< 0.001).

**Fig. 3.** Basin-scale relationships between macrofauna biodiversity and ecosystem functioning and efficiency. (a) Relationship between expected species richness (ES$_{(50)}$) and ecosystem functioning, expressed as total benthic biomass (mgC/m$^2$). The equation of the fitting line is $y = x^{1.43}$ (N = 27; $R^2 = 0.32$; P< 0.01). (b) Relationship between expected species richness (ES$_{(50)}$) and ecosystem efficiency, expressed as macrobenthic biomass to prokaryotic biomass (MBM : TPB). The equation of the fitting line is $y = e^{(-1.90+0.12x)}$ (N= 27; $R^2 = 0.33$; p< 0.01 ).

**Fig. 4.** Basin-scale relationships between macrofauna functional diversity and ecosystem functioning. Relationship between functional diversity, expressed as expected richness of deposit feeders (EDF$_{(30)}$) and expected predator richness (EPR$_{(20)}$), and ecosystem functioning, expressed as total benthic biomass (mgC/m$^2$). The equations of the fitting line are respectively (a) $y = e^{(6.67-4.83x)}$ (N= 9; $R^2 = 0.98$; p< 0.01) and (b) $y = x^{2.71}$ (N= 9; $R^2 = 0.61$; p< 0.05) in the Atlantic Ocean and (c) $y = e^{(-1.60+2.82x)}$ (N= 27; $R^2 = 0.98$; p< 0.01 ) in the Western Mediterranean basin.



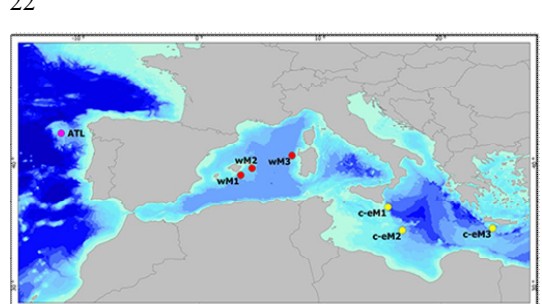


**Fig. 1.**

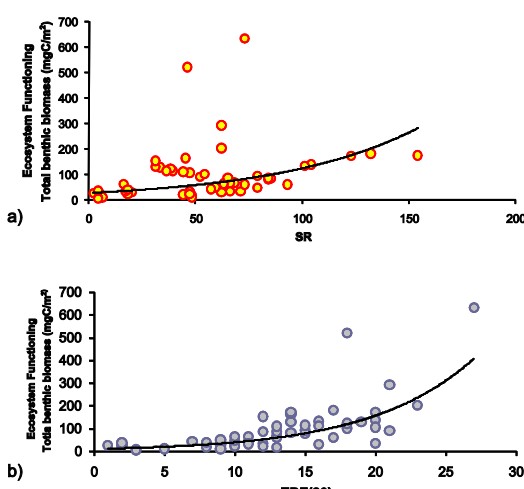


**Fig. 2.**

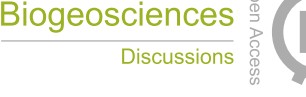

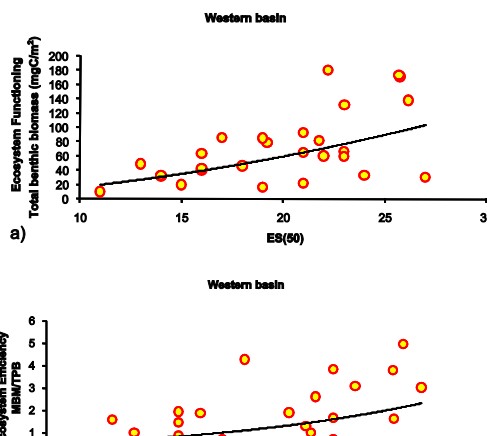


**Fig. 3**.

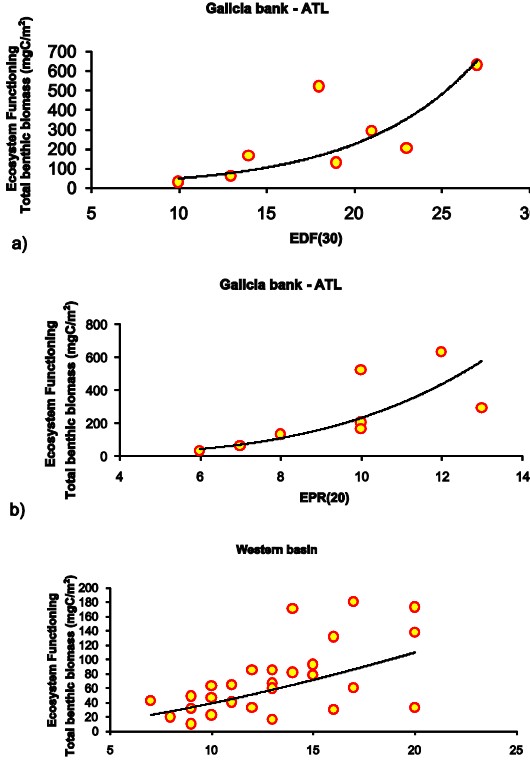


**Fig. 4.**