# Peer review of "Deep-sea ecosystem: a world of positive biodiversity – ecosystem functioning relationships?"

_Biogeosciences, 2016_

## Referee Comment (RC1) · Anonymous Referee #1 · 15 Feb 2016

General comments:

The analyses are very complicated and could be more focused. It looks like the authors were trying very hard to find a significant relationship and managed to find a few after trying different ways of quantifying diversity and function. The results of the analyses are not convincing me that there exists a positive relationship, and I find the conclusions much too strong given the few relatively weak correlations that were found. The authors should decide on the most meaningful measures of diversity and function a priory, then stick to them in their analyses. The authors need to include abundance in their analyses. More details are needed in the Methods in order to understand how they conducted their sampling and statistical analyses. the results are difficult to follow and

need simplified. Also this preference for using exponential curves rather than linear one (when in most cases they fit about equally well, and AIC values probably are only larger for exponential fits than linear ones by very small margin) just perpetuates this over-complicated view of BEF relationships in the deep sea.

Detailed comments: Line 89: Gagic et al. not an appropriate reference for this statement Line 97: what does the 7-9 refer to? Line 100: it is unclear what you meant by complex systems Line 105: how do the results of experimental and field studies differ, more specifically? Line 111: It is very debatable whether the deep sea is the "most diversified environement"! Remove. Line 112: Ingels & vanreusel is not a relevant reference for this statement. Find better ones. Line 119: mutualistic interactions have by no means been "shown". They have been suggested as a possible mechanism to explain the pattern. Nothing is known about mutualism, competition, etc... Line 121: Danovaro et al. 208 do not demonstrate anything about the effect of species loss. Their correlative study only suggests that species loss may be associated with loss of function. Line 124: what is this (8)? Line 129: all deep-sea studies are correlative. Therefore they do not investigate the role of fauna/microbes in "enhancing" function. One needs to be very careful about how these studies are referred to. Lines 138-140: there is nothing on the Introduction about these last 2 objectives. These topics need to be covered in the Introduction.

Line 174: so to recapitulate, there were 7 areas x 3 depths x 3 stations x 3 replicates = 81 core samples each from independent boxcore deployment? Line 189: explain what you mean by sensu lato. It's an important detail needing to be spelled out here. So the macrofauna includes all the fauna retained on the 300 micron sieve (including nematodes etc..)? Also need to clarify whether macrofauna were sampled from the sediment remaining after meiofauna and sediment subcoring (what was the surface area of the macrofauna samples?). It is not clear in the text as it is now. Lines 194-199: we need to know whether these data are new or have been published already in Baldrighi et al. 2014. Wont total biomass be strongly dominated by bacteria? Lines

[Figure]

Line 206: Did you always have at least 50 individuals? Or were individual boxcores pooled for statistical analyses. We need more details of this for all the analyses, i.e., what is a sample- a boxcore or three boxcores pooled together? 204-205: it is not clear what the difference between species richness and total number of species is! Line 210: what were the feeding groups? need a reference for this. Line 212: need more details of how bioturbation potential was estimated. Line 225: secondary production, not production of renewable resources. Line 241: BEF relationships can be affected by the... Line 258: I dont understand. How were they used to account for environmental effects exactly? Line 260: before testing for the effects of what? diversity? And how do you make non-linear fits in the DistLMs?

You really need to investigate the effects of faunal abundance on your measures of function. I have the feeling that some sites have high species richness simply because they have higher abundance, and therefore have also higher biomass. It seems circular. If abundance is included, I think relationships might well disappear.

There are quite a few measures of function and several measures of diversity (and therefore many potential combinations of predictor and responses) so it is not surprising that some relationships will come out significant.

It would be better to show partial regression plots so as to better reflect the relationship after the covariates have been accounted for.

Table S5: why not just give P values for before and after analyses? I see that none of the relationships are significant after covariates are accounted for (and R2 values are low). Second line of caption: i do not understand second sentence.

Table S4: re-define what you mean by large-scale analysis. Independent, not indipendent. Shouldnt we see the same relationships shown in Table S4 again in table S5a? Why do degrees of freedom vary between the different analyses? Did you check that you covariates or not strong collinear?
[Figure]

I did not comment on the rest of the Discussion because I think there needs to be some major changes made to the Methods and Results sections, which would require some major re-working of the Discussion as well.

Looking at figure 3 and 4, it is clear that a linear relationship would be just as justified as an exponential one! The simplest explanation is always best, even if AIC values might differ by small amounts. This whole mystery about exponential BEF relationship in the deep sea arose because some researchers got slightly better fits by using exponential curves than linear ones, not because the relationships are actually exponential.

---

## Referee Comment (RC2) · Anonymous Referee #2 · 29 Feb 2016

1. General comments The authors aim to analyze the relationship of macrofauna biodiversity and ecosystem functioning on deep-sea slopes and the potential influence of rare species in this relationship. Partly they use hypotheses and methods previously used by the authors themselves or other (e.g. Danovaro et al. 2008, 2012). New is the focus on the macrofauna size class and the very interesting investigation of the role of rare species in the BEF relationship in the deep sea. The dataset used in this study appears to have been used and published already several times by the authors (e.g. Baldrighi et al. 2014, Baldrighi & Manini 2015). As macrofauna datasets from deep-sea ecosystems are generally rare, using existing datasets repeatedly in different approaches is not at all a flaw. However, if this work is to be accepted as stand-alone

work, the authors should emphasize more on what is new in this manuscript compared to what was done in previous work. The introduction should be more structured and include definitions of the central terms and concepts used in the manuscript. The methods have to be explained more in detail to give the reader a chance to understand what was done and how it was done. The environmental differences of the sampled regions have to be explained more detailed, also how the influence of the different environmental parameters was tested has to be explained more clearly. In the light of the different sample size, the use of other taxonomic diversity measures than ES(n) should be reconsidered. Information about the estimated abundances should be given. The authors should explain clearly why their approach of using only single traits as proxy for overall ecosystem functioning or trophic diversity as proxy for overall functional diversity is appropriate, especially regarding the presented results and in the light of other publications that use a more holistic approach (e.g. work of Julie Bremner or Stefan Bolam).

2. Specific comments Abstract 53 The deep sea, not the deep-sea floor is the largest biome on earth. 57 What about previous work including macrofauna (Danovaro et al. 2012, Baldrighi & Manini 2015)? Some information about the used method should be included in the abstract.

Introduction The introduction could be better structured and longer, more references could be given. Definitions and explanations of the major terms and concepts should be provided (traits, ecosystem functioning, efficiency, . . .). Briefly the general attributes of benthic deep-sea ecosystems should be described. The question about the role of rare species is the most interesting in this manuscript, some information about rare species should be given (currently they show up for the first time in hypothesis 3, line 139). 97 Give references for these BEF studies in deep sea. What is 7-9? 100-101 Provide a reference for a study with animals. 103 Gagic et al. 2015 could be added as reference here. 105 Provide and additional Ref. to Lefcheck and Duffy 2014 here. Also maybe use 2015 instead of 2014 (Ecology Ref.) 109 ff More information

about the ecosystem functioning of deep-sea ecosystems should be given, not only bioturbation. See e.g. Thurber et al. 2014. 121 Delete Danovaro reference here as it is given already at the begin of the sentence. 123 "... in relation to the functional traits and the species involved.." This is not clear to me, rephrase. What about mentioning environmental factors here? Also functional traits are mentioned here the first time and not explained before. 124 "study 8"? Give a reference. 136 If "The observational – correlative approach" is a known procedure give a reference, otherwise I would use "Here we use a observational-correlative" approach, or similar. 136 Delete "the truth of" 138 Delete the reference here, rather include in previous part, e.g. line 103. 139 Rare species as typical feature of deep-sea benthos should be mentioned and explained already before, so the info and reference could be removed here.

Methods The different regions and environments should be explained. Currently the Atlantic station is described more detailed than the other stations. 144 Selected based on which criteria? Why the different depth zones, if they are not used (e.g. for comparison between the 7 shallowest, intermediate, and deepest stations)? 149 If you refer to Fig. 1 regarding depth, the Fig. should include a color code for water depth. 149 Change to "range from 5 to 30 cm. . ." 154 Explain "well-established trophic difference" or rephrase. The paragraphs 2.2 – 2.4 are not clear. Also better combine them in one paragraph named e.g. "Sampling". Information about study area should be moved to the previous paragraph study area. 162 A reference for BIOFUN? 166 ff This info should go to "study area". What means "topographically regular"? If there are diff. conditions, they should be described in the section "study area". Why here table S1? 178 f Where the subsamples taken from the macrofauna cores? Was the removed area subtracted for the calculations per m2? Or was there one core used separately for the subsamples? 189 What is senu lato here? 194 ff The methods should be explained here, the readers can not be expected to read 3 other publications of the authors to understand the methods of the present paper. 2.5 Functional diversity is also a type of biodiversity. Maybe use "analysis of diversity" or "taxonomic and functional diversity of macrofauna". 202 This has to be explained, e.g. which traits were used? 204

How was dealt with the different sample area of the cores when assessing species richness? This should be explained here. 206 ff Rarefaction is sensitive to low abundances, this should be brought up here. Deep-sea macrofauna samples often show very low abundances, even below 50 individuals per m2. Provide an abundance table (individuals/m2) or ranges of abundance to show that potentially too low abundances are not an issue here. Also provide more references than Danovaro 2008 to underline that the measures of diversity used here are appropriate with this kind of dataset. -> as the authors state that rarefaction/ES(50) is the best approach for samples with different sample sizes, why are the other diversity measures still used? It would also reduce the number of tests and clear the results if only ES(50) was used. 208 The Definition of functional diversity should be in the introduction. Also it could be broader, see e.g. Petchey & Gaston (2006, Functional diversity: back to basics and looking forward). Give more references. 209 ff The authors state before that functional diversity is "the range of functions performed by organisms in an ecosystem", but here focus solely on trophic diversity. It should be explained clearly why trophic diversity can used as proxy for the overall functional diversity of macrofauna. And be aware of papers e.g. Bremner et al. (2003) Assessing functional diversity in marine benthic ecosystems: a comparison of approaches 211 Expected numbers of e.g. deposit feeder EDF30 -> have there been enough individuals & diff. feeding types for "30"? See before – better provide an abundance table. How is biomass converted to carbon? If published conversion factors are used, give references. 225 Secondary production is the measure of renewable resources by an ecosystem. Also the reference given here – Rowe et al. 2008 – work with secondary production. This has to be rephrased. 241 "BEF relations can be determined by the effect of the spatial scale of investigation..." this is not clear to me, rephrase. Wouldn't it be interesting to also compare the BEF relationships in groups of depth zones? 243 "Large spatial scale" – this sounds like a large, connected sampled area, like in a monitoring program, not like seven very separated sample stations. Maybe just use "entire dataset". 247 This is not clear to me: It sounds like you analyze the relationship between BEF (which is the relationship of biodiversity and ecosystem

functioning) and efficiency. From S4 I understand that you tested all diversity measures separately with efficiency in the three models. Rephrase to make this clear to the reader. The DISTLM approach to test for environmental effects is not clear to me, this should be explained. The method section contains no information what was done with the rare species.

Results & Discussion It would be helpful to have the three hypotheses from the introduction as headlines to orientate in the discussion. Or, alternatively, the hypotheses could be formulated to fit to the large-scale and basin-scale approach. The authors underline the important effect of the different environmental conditions on the biodiversity-ecosystem functioning relationship they observe in this study. More information like the Kröncke et al. (2003) reference should be provided to show how biodiversity of macro-fauna was previously described in the Mediterranean, and which environmental parameters are known to haven a positive or negative effect (in the Mediterranean and in general). This would enable the reader to position these novel findings in the frame of existing knowledge. Also the reasonability of pooling data that are geographically dispersed as in the present study (i.e. the "large-scale" approach) should be discussed. 266 Give a reference for these statements. 275 f "existence of a BEF relationship appeared to be closely linked to the diversity and ecosystem functioning measures used"? Do the authors still think biomass an appropriate measure for functioning? Or trophic diversity an appropriate proxy for functional diversity? 284 "Positive relationship of diversity and efficiency"? In line 299 the authors state that there is no significant relationship. 308 ff What about dwarfism in the deep-sea in general (see Gage & Tyler 1991)? Moreover, many deep-sea predators are very mobile and therefor not included in classical macrofauna sampling. 319 This should be explained. 321-323 Also, explain to make clear. 331 Finally here at the end of the first section of the discussion the authors refer to their first hypothesis, which could not be confirmed with this study. The authors conclude that "this suggests that they may not encompass the full array of key macrobenthic functional traits that underpin ecosystem functioning". I think this is one main outcome of this study and should be discussed more detailed, more references

from literature should be given in which more functional traits were used (see again e.g. Bremner papers, or Bolam). The sentence about isotope studies seems a bit lost here. 336 – 338 Explain how and why on base of your results. The environmental gradients in the study area or effects of environmental parameters were not described or discussed at all so far. 344 These different environments should be described in the methods section. 358-360 This is too general. 370 ff A turnover in species composition must not lead to a change in the functional structure per se, so I suggest to delete the example in the brackets. The big strength of the functional trait approach is, that it can be applied to study changes in function over large spatial scales, regardless of potential changes in community composition. 380-383 General info of rare species, along with a brief characterization of deep-sea ecosystems should be given in the introduction. Also it would be interesting to have a total species list provided as supplement, with rare species marked. 423 The deepest sample station in this study is 3068m. Do the authors expect that the number of rare species might be higher in deeper areas? What did other studies find? 445-446 Why? The result from this study is quite clear. 446-447 Remove, this is not a conclusion.

3. Technical corrections 129 Change to Amaro et al. 2010. 134 In reference list it is Gamfeldt et al. 2015, in manuscript 2014. 278 ".. index SHOWING a positive relationship with..." 386 In the present study we define, or the present study defines... 545 The reference Frid et al. 2015 is not in the Manuscript. S3 and S6 Typo: "dependent" variable

---

## Author Comment (AC1) · 21 Mar 2016

Dear Reviewer 1, We would like to thank you for taking the time and effort in reviewing our manuscript. You have brought forward several issues that need clarification, some of which require changes in the manuscript. We have carefully reviewed on the basis of your suggesting and we have clarified our point of view with changes and proposed solution. We hope that all suggestions, comment and requests have been carefully considered and reported in the amended version of the ms.

Please find in Supplement the Authors' reply and the MS with authors' changes.

Please also note the supplement to this comment:

http://www.biogeosciences-discuss.net/bg-2016-26/bg-2016-26-AC1-supplement.pdf

---

## Author Comment (AC2) · 21 Mar 2016

Dear Reviewer 2,

We would like to thank you for taking the time and effort in reviewing our manuscript. You have brought forward several issues that need clarification, some of which require changes in the manuscript. We have taken everything on board and hope that the changes we are suggesting are sufficient. In cases where we thought there may have been confusion of misinterpretation, we have clarified our point of view and have proposed a solution. We hope that you may receive the proposed revision positively.

Please find in Supplement the Authors' reply and the MS authors' changes.

[Figure]

Please also note the supplement to this comment:
http://www.biogeosciences-discuss.net/bg-2016-26/bg-2016-26-AC2-supplement.pdf
* * *